# Quantitative Analysis of Hair Luster in a Novel Ultraviolet-Irradiated Mouse Model

**DOI:** 10.3390/ijms25031885

**Published:** 2024-02-04

**Authors:** Kyung Bae Chung, Young In Lee, Yoo Jin Kim, Hyeon Ah Do, Jangmi Suk, Inhee Jung, Do-Young Kim, Ju Hee Lee

**Affiliations:** 1Department of Dermatology and Cutaneous Biology Research Institute, Yonsei University College of Medicine, Seoul 03722, Republic of Korea; chungkyungbae@yuhs.ac (K.B.C.); ylee1124@yuhs.ac (Y.I.L.); 2Scar Laser and Plastic Surgery Center, Yonsei Cancer Hospital, Seoul 03722, Republic of Korea; 3Global Medical Research Center Co., Ltd., Seoul 06526, Republic of Korea; kyj@gmrc.co.kr (Y.J.K.); dha0201@gmrc.co.kr (H.A.D.); rose@gmrc.co.kr (J.S.); ihjung@gmrc.co.kr (I.J.)

**Keywords:** hair luster, hair gloss, hair shaft, ultraviolet radiation, mouse model, minoxidil

## Abstract

Hair luster is a key attribute of healthy hair and a crucial aspect of cosmetic appeal, reflecting the overall health and vitality of hair. Despite its significance, the advancement of therapeutic strategies for hair luster enhancement have been limited due to the absence of an effective experimental model. This study aimed to establish a novel animal model to assess hair gloss, employing ultraviolet (UV) irradiation on C57BL/6 mice. Specifically, UVB irradiation was meticulously applied to the shaved skin of these mice, simulating conditions that typically lead to hair luster loss in humans. The regrowth and characteristics of the hair were evaluated using a dual approach: an Investigator’s Global Assessment (IGA) scale for subjective assessment and an image-based pixel-count method for objective quantification. These methods provided a comprehensive understanding of the changes in hair quality post-irradiation. To explore the potential reversibility of hair luster changes, oral minoxidil was administered, a treatment known for its effects on hair growth and texture. Further, to gain insights into the underlying biological mechanisms, bulk RNA transcriptomic analysis of skin tissue was conducted. This analysis revealed significant alterations in the expression of keratin-associated protein (KRTAP) genes, suggesting modifications in hair keratin crosslinking due to UV exposure. These changes are crucial in understanding the molecular dynamics affecting hair luster. The development of this new mouse model is a significant advancement in hair care research. It not only facilitates the evaluation of hair luster in a controlled setting but also opens avenues for the research and development of innovative therapeutic strategies. This model holds promise for the formulation of more effective hair care products and treatments, potentially revolutionizing the approach towards managing and enhancing hair luster.

## 1. Introduction

Hair luster is considered a critical element of aesthetic significance from the perspective of beauty and health. This attribute is primarily determined by the condition of the hair cuticle, the outermost layer of the hair shaft [1]. A smooth and intact cuticle is key to facilitating specular reflection, which gives hair its characteristic glossy appearance. However, maintaining hair luster can be challenging, as it is frequently compromised by a variety of stressors. These include exposure to ultraviolet (UV) radiation, physical and chemical damage, and nutritional deficiencies, all of which can degrade the hair shaft and cuticles, diminishing the hair’s natural shine and smoothness [2,3].

The available solutions for managing hair gloss are predominantly topical, such as conditioners and serums. While these products offer temporary improvement by smoothing the cuticle surface, they do not influence the properties of the hair shaft as it grows. This limitation underlines the need for more systemic approaches, such as nutraceuticals or pharmaceutical interventions, that could potentially alter the intrinsic characteristics of hair during its regenerative cycle. However, a major obstacle in developing such treatments is the lack of an established in vivo model to study and measure the effects of these interventions on hair luster.

This gap in research and development has led to an influx of various nutraceuticals in the market, many of which have not been thoroughly tested or proven effective. Recognizing this need, our study introduces a novel animal model designed to simulate hair luster loss through controlled UV irradiation. This innovative approach aims to mimic the environmental and physiological conditions that contribute to the degradation of hair luster in humans. By using this model, we can effectively evaluate the potential of systemic strategies aimed at enhancing hair gloss. This research not only addresses a significant void in cosmetic and dermatological studies but also paves the way for more rigorous testing and development of hair care products and treatments, ultimately contributing to a better understanding and maintenance of hair health and aesthetics.

## 2. Results

### 2.1. Induction of Hair Roughness Using UVB Irradiation

UVB irradiation of depilated skin induced the regrowth of rough hair at all tested UV dosages (Figure 1a). Notably, the doses of 80 mJ/cm^2^ and 100 mJ/cm^2^ resulted in the formation of alopecic patches. Due to these effects, subsequent analyses focused on mice irradiated with a dose of 60 mJ/cm^2^ or lower. In the group irradiated with 60 mJ/cm^2^, a loss of hair luster by −1.833 (±0.289) points occurred and was more pronounced compared to that in the 40 mJ/cm^2^ group (−1.111 (±0.192) points) and the 0 mJ/cm^2^ control group (−0.333 (±0.726) points) as determined by the Investigator’s Global Assessment (IGA) scale (Figure 1b). Pixel-based gloss assessment also indicated a loss of hair luster in the 60 mJ/cm^2^ group (102.77 (±7.601) pixels) compared to the 0 mJ/cm^2^ (251.17 (±18.315) pixels) and 40 mJ/cm^2^ groups (222.33 (±46.810) pixels) (Figure 1c). A statistical correlation (R^2^ = 0.998, *p*-value = 0.029) was found between the IGA scale and the pixel-based assessment (Figure 1d).

### 2.2. Hair Roughness following UVB Irradiation Can Be Modified

Hair roughness induced by UVB irradiation (0 mJ/cm^2^, 60 mJ/cm^2^) was treated with vehicle or oral minoxidil (0.5 mg/kg) (Figure 2a). Among the UV-irradiated group (60 mJ/cm^2^), oral minoxidil administration reversed the loss of hair luster from −1.833 (±0.289) points to 0.389 points (±0.631) (Figure 2a,b). Pixel-based gloss assessment also indicated a notable improvement after minoxidil treatment (341.47 (±35.068) pixels) compared to the vehicle group (102.77 (±7.601) pixels) (Figure 2c). Also, in the UV non-irradiated group (0 mJ/cm^2^), mice treated with oral minoxidil showed improved hair gloss, showing 1.157 (±0.167) compared to 0 mJ/cm^2^ + vehicle group (−0.333 (±0.726)) on the IGA scale. Similarly, pixel-based gloss assessment yielded similar results. The results showed a high statistical correlation (R^2^ = 0.914, *p*-value = 0.002) between visual and pixel-based evaluations (Figure 2d). The minoxidil-treated group exhibited a higher growth rate than the saline-treated group; however, on day 21, no significant difference in growth rate was observed across the different UV dosages. (Appendix A).

### 2.3. Gene Expression Profiling Identifies Increased Genes Associated with Keratinization

To elucidate the mechanism underlying the loss of luster in this model, irradiated skin was investigated using bulk RNA sequencing, with a focus on comparing the group irradiated with 60 mJ/cm^2^ and nonirradiated control group. Principal component analysis (PCA) was performed on three samples each from the control group and the group irradiated with 60 mJ/cm^2^, revealing distinct transcriptomic differences (Figure 3a). An unsupervised clustering heatmap demonstrated the gene sets that were significantly different between the two groups (Figure 3b). Notably, among the 106 upregulated genes (Log2FC > 1.0, adjusted *p* < 0.05) in skin samples exhibiting lost hair luster, 30 (28.3%) were associated with keratinization (Figure 3c,d). The genes upregulated in the UV-irradiated group included *Krtap9-1*, *Krtap5-4*, *Krtap4-13*, *2310061N02Rik* and *Krtap5-5*, which are recognized as keratin-associated proteins (KRTAPs). This indicates a potentially abnormal keratinization process that may lead to changes in hair shaft characteristics, thereby contributing to the loss of hair gloss.

## 3. Discussion

Despite the significant interest in “hair gloss”, there has been a limitation due to the lack of a definitive animal model for evaluating novel therapeutics improving hair luster. Therefore, we developed a novel animal model that induces hair roughness leading to the loss of hair luster. Among various factors that cause hair coarseness, UVB radiation was selected in our model. UVB, a component of solar radiation, can be absorbed by hair fibers [2,4]. This absorption leads to breakage of disulfide bonds, formation of reactive oxygen species (ROS), and ultimately resulting in the photodegradation of hair [5]. Additionally, UVB exposure may induce apoptotic cell death and inhibit hair growth [2]. Thus, UVB can serve as a useful tool for modulating hair gloss. With increasing UVB irradiation up to 60 mJ/m^2^, the hair gloss decreased. At doses of ≥80 mJ/cm^2^, erythematous patches with slight vesicular formation and multiple scarred alopecic patches were identified, suggesting a third-degree burn. Irradiation with UVB at doses ranging from 0 to 60 mJ/m^2^ led to a dose-dependent decrease in hair gloss, with 60 mJ/cm^2^ being the most effective and consistent dose in repeated experiments.

We also proposed a hair gloss scale as an indicator to evaluate this model. As hair gloss is typically recognized by the naked eye as a specular reflection of light, our model focused on evaluating hair gloss using the IGA scale. Additionally, the significant correlation of the IGA scale with an objective, quantitative, photo-based pixel analysis suggests that IGA is an easy and practical tool for assessing hair luster in this model. Similarly, a three-dimensional digital microscope was shown in a previous study to be an alternative tool for evaluating hair luster, correlating well with a naked-eye assessment [6]. Though our study did not evaluate hair shaft at a microscopic level, the naked-eye approach which may be the golden criteria for hair gloss changes, has been fully proven by two assessment methods. These two methods are generally easier, and more comfortable methods for evaluation of hair gloss. Further studies are required to evaluate the macroscopic changes caused by UVB and its relationship of the microscopic level using single hair fibers.

Though the mechanism of UVB on hair shaft is diverse, the definitive cause of loss of hair luster by UVB in regrowing hair follicles is currently not known. Some studies have shown that with regard to the hair cycle, UVB can induce early catagen and cause oxidative damage in human hair follicles [7]. However, in mice, UVB up-regulated the expression of the Wnt signaling pathway and prolonged anagen and telogen phases. These data suggest that the direct effect of UV on hair growth is not clear [8]. In our study, regarding that hair growth was fully grown in D21 may indirectly suggest that no definite change in hair cycle was caused by UVB. Rather, UVB can affect the hair follicle without modulating the hair cycle in such ways as the formation of reactive oxygen species or apoptosis. Thus, this study explored the mechanisms underlying changes in hair luster due to UV irradiation by analyzing transcriptomic alterations via bulk RNA sequencing. The analysis revealed distinct transcriptomic differences in the UVB irradiated group, notably in genes associated with hair keratinization, including significant changes in *Krtap* genes. KRTAPs, crucial in hair shaft development, contribute to hair characteristics by forming a matrix between keratin filaments creating disulfide bonds [9,10,11]. Historically, evolutionary changes in KRTAP have led to diverse hair characteristics among different animal species, suggesting the importance of change caused by KRTAP [12,13,14]. Also, upregulation of *Krtap* genes is linked to asymmetrical hair growth leading to loss of hair luster or change in fur characteristics [10,15]. Stamatas et al. [16] had shown that when minoxidil is treated in humans, KRTAP gene levels change. This can also indirectly suggest the molecular changes in KRTAP levels can be induced by minoxidil and affect hair shaft properties including an improvement in hair gloss. Still, whether KRTAP up or down regulation cause changes in hair luster should be further studied. Regarding previous studies, the balance of KRTAP across the hair shaft seems to be crucial, however more molecular studies are required. Moreover, varying expression patterns of KRTAP across different anatomical locations within follicular structures have been observed [17]. This underscores the necessity for detailed spatial profiling of each upregulated KRTAP at the subfollicular level in future research. Although our study does not completely elucidate the mechanism, it successfully established a novel and practical model for influencing and potentially reversing hair luster loss through therapeutic interventions.

To determine whether the change in hair roughness was reversible, minoxidil (0.5 mg/kg) was administered orally for 2 weeks following UV irradiation in our model. Notably, this approach demonstrated that hair gloss was reversible with therapeutic intervention. Given that the assessment of novel treatment modalities necessitates an examination of their capability to reverse the loss of hair luster, this animal model, which exhibited an improvement in hair luster with minoxidil, can be deemed an appropriate model. Also, minoxidil may serve as a benchmark for the development of dietary products and medications aimed at enhancing hair luster. Further development of novel therapeutics including pharmaceuticals and nutraceuticals can be used in our model. Nevertheless, an in-depth investigation into the precise mechanism of minoxidil on hair luster is imperative. While minoxidil is recognized for its ability to increase hair diameter and induce or prolong anagen, its correlation with hair luster remains unidentified. Noteworthy is the fact that patients treated with oral minoxidil often report complaints of poor hair texture or alterations in the curliness of hair, suggesting that minoxidil may influence not only hair diameter and cycle but potentially also hair shaft properties [18]. Our data indicate that minoxidil-treated mice exhibited an increased hair growth rate at D14, potentially leading to rapid hair growth. However, the dependency of hair luster on hair growth rate is ambiguous, and further studies, particularly focused on hair shaft quality, are warranted. Further transcriptomic evaluation of minoxidil induced improvement of hair gloss may shed light on understanding the pathogenesis.

Therefore, this study is not without limitations. Firstly, our study has limitations in specific mechanisms of UV-induced hair luster changes. Since the role of UVB on hair follicles is still an ongoing topic, further studies are required to identify UVB effects on hair follicle epithelial cells and the changes in newly growing hair. Secondly, the mechanism of minoxidil affecting hair luster needs to be further studied. The changes in hair curl and other characteristics observed in some patients using minoxidil indicate the need for further research to understand its influence on hair shaft curvature and surface cuticle modifications that lead to changes in luster. Finally, several KRTAPs which were identified in transcriptomic changes in RNA sequencing data need further evaluation. Though KRTAP is known to affect hair shaft characteristics and is a very specific protein in hair shafts. Transcriptomic changes in KRTAPs indicate that UVB-induced alterations in hair gloss are highly specific to hair luster, with only minimal changes observed in immune responses, apoptosis, or metabolic gene expression at a non-burn-inducing dose. Further cellular level evaluation on KRTAP changes in the novel mouse model is required.

Still, this model has strength in that it is a novel mouse model that can induce loss of hair gloss using a simple way of UVB irradiation. Evaluation by IGA and a pixel-based model with high correlation has suggested a novel scoring system for hair luster. Also, the reversibility of hair gloss can provide chances of developing therapeutics for hair gloss. Finally, transcriptomic changes with minimal involvement in inflammation nor apoptosis suggest that KRTAP gene changes are caused by UVB, and the irradiation only can affect the hair shaft. This model is important in that it is one of the novel models of hair shaft change induction. This study mainly focused on the macroscopic level of hair luster, however as mentioned, many studies can be followed including hair diameter, length, and curvature. Additionally, our model is versatile in analyzing popular therapeutic methods that lack substantial evidence.

## 4. Materials and Methods

### 4.1. Mice

This study utilized 5-week-old male C57BL/6 mice (Orient Bio, Seongnam, Republic of Korea) as experimental animals. Utilizing 5-week-old mice in studies serves as an efficient approach to synchronize hair follicle cycles, effectively minimizing variations caused by different hair cycle stages. All mice were group-housed, with 3–5 mice per cage, provided with water and standard chow, under a 12-h light/dark cycle, in accordance with guidelines for the Association for Assessment and Accreditation of Laboratory Animal Care International. The mice were allowed free access to food and water and were given 7 days to acclimatize to their environment before starting the experiment. Sedation for mouse was done using inhalant anesthetic (isoflurane 1.5–2%) (N01AB06, Hana Pharm, Seoul, Republic of Korea) when required for shaving of hair and UVB irradiation. The study protocol was approved by The Institutional Animal Care and Use Committee of Yonsei University and experiments were performed in accordance with the AAALAC International guidelines (IACUC No. 2022-0267).

### 4.2. UVB Irradiation on Naked Mouse Skin

The mice were randomly assigned to different groups and carefully prepared by shaving hair using hair clippers from their backs to ensure that no skin trauma occurred during the process. To induce roughness in newly growing hair, UVB irradiation was applied using a 312-nm lamp (BLX312 UV Crosslinker; Vilber Lourmat, Marne-la-Vallée, France) after sedation of mice. The dosages varied from 0 to 150 mJ/cm^2^ per exposure (0, 40, 60, 80, and 150 mJ/cm^2^) and were administered over a period of 5 days (Figure 4a). Irradiation was repeatedly administered for consecutive 5 days with a 24-h interval. The total accumulative irradiation dose was each 0, 200, 300, 400, 600 mJ/cm^2^.

### 4.3. Evaluation of Hair Regrowth Pattern

After the initiation of irradiation, hair regrowth on the mouse skin was monitored every 7 days (D7, D14, and D21). A comprehensive evaluation of the regrown hair luster was conducted on day 21, which was 16 days after the final irradiation session (Figure 4a). Hair regrowth percentage was evaluated in terms of hair growth area by visual assessment (Appendix A). Hair regrowth area was evaluated every D7, D14, and D21.

### 4.4. Sacrification of Mouse and Skin Strip

To assess hair gloss, back skin samples were collected after mice sacrification. Mice were euthanized inside a chamber gradually filled using carbon dioxide (CO_2_) with at a rate of 30–70% volume per minute and in accordance with the AAALAC International guidelines. The obtained back skin included epidermis, dermis and subcutaneous fat tissue while preserving the hair.

### 4.5. Photograph

The skin samples were meticulously prepared and positioned on a cylindrical bar (60 mm in diameter), ensuring consistent exposure to the light source. The individual light source was carefully calibrated to maintain a fixed intensity of 650 Lux. The camera used for capturing images was strategically placed at a distance of 40 cm from the sample, forming a 70-degree angle with the LED light source to achieve optimal lighting conditions. (Figure 4b). Photographic parameters were rigorously controlled; the camera was set with a shutter speed of 1/125, an aperture of f/5.6, International Organization for Standardization(ISO) sensitivity 200, and an F-number of 10, to ensure high-quality, consistent images. To achieve objective quantification of hair luster, the photographs, containing only the hairy parts, were uniformly cropped to a size of 1152 × 868 pixels and subsequently analyzed. This uniformity was critical for the subsequent image analysis. The cropped images were then subjected to a detailed analysis process, which allowed for precise and objective quantification of hair luster, providing valuable data for the study.

### 4.6. Assessment of Hair Luster by Investigator’s Global Assessment (IGA) Scale

Hair gloss was evaluated by three independent investigators using the investigator’s global assessment (IGA), a 5-point assessment tool. The scale categorizes hair condition into five levels based on visual inspection: −2, extremely rough; −1, rough; 0, normal; 1, glossy; and 2, extremely glossy (Figure 4c).

### 4.7. Assessment of Hair Luster by Computational Method

For the objective quantification of hair luster, the photographs were analyzed using a computational program designed to count the luminous pixels (Figure 4d). Cropped photographs of the same size were evaluated using the I-MAXPLUS software (v 1.0, ING plus, Seoul, Korea). The software identified and quantified the shiny areas of the hair in photographs, calculated both the area and intensity of luster, and presented these data as pixel values. The final objective luster grades were classified into five categories aligned with the investigator’s global assessment (IGA) scale.

### 4.8. RNA Sequencing and Analysis

A 5-mm-sized skin sample was obtained for bulk RNA sequencing analysis. Total RNA from the skin tissues was extracted using TRIzol™ reagent (Invitrogen, Carlsbad, CA, USA). The extracted total RNA, oligo dT primers, and Accupower RT Premix (Bioneer, Daejeon, Korea) were used for cDNA synthesis. Subsequently, synthesized cDNA was sequenced using an Illumina NovaSeq 6000 next-generation sequencing (NGS; Illumina, San Diego, CA, USA) platform. DESeq2 was used for analysis of bulk RNA sequencing. A total of 13,133 genes were included in the analysis removing low expressing genes. Differentially expressed genes (DEG), were obtained using DESeq2 and were then classified using the Metascape pathway analysis tool (Log2FC > 1.0, adjusted *p* < 0.05) [19,20].

### 4.9. Oral Minoxidil Administration in Modification of Hair Gloss

Minoxidil (Sigma-Aldrich, St. Louis, MO, USA) was administered by oral gavage at a dose of 0.5 mg/kg every 24 h, while the control group was treated with saline of the same volume once a day for 2 weeks.

### 4.10. Statistical Analysis

All experimental data were presented as a mean ± standard deviation or percentages, and all experiments were conducted in triplicate (*n* ≥ 3). Statistical analysis was performed using the SPSS Statistics 25.0 (IBM Corp., Armonk, NY, USA) program. Statistical significance was calculated using the independent samples *t*-test. Correlation was evaluated by Pearson’s correlation. Statistical significance was considered at *p* < 0.05 and *p* < 0.01, indicated by * and **, respectively, depending on the group of comparison.

## 5. Conclusions

This study successfully addressed the challenge of lacking a definitive animal model for evaluating therapies targeting hair luster. The novel animal model, established through controlled UV irradiation, provided practical insights into systemic strategies for enhancing hair luster. UVB irradiation induced dose-dependent hair roughness, with 60 mJ/cm^2^ as the most effective dose. The model, validated by Investigator’s Global Assessment (IGA) and pixel-based evaluations, facilitated a clear understanding of the impact of UVB on hair gloss. Furthermore, the reversibility of UVB-induced hair luster loss was demonstrated through oral minoxidil administration, emphasizing its potential as a benchmark for developing products aimed at enhancing hair luster. The study’s exploration of transcriptomic differences, particularly upregulated keratin-associated proteins (KRTAPs), provides valuable insights for future research on understanding and potentially reversing hair luster loss through therapeutic interventions. Our study is novel in that the animal model is a novel model leading to loss of hair gloss, evaluated by pixel-based and IGA-based methods. Also, the possible role of KRTAP and hair gloss is suggested. This model would shed light on the understanding of hair luster and the development of diverse pharmaceutical and nutraceutical treatment methodologies.

## Figures and Tables

**Figure 1 ijms-25-01885-f001:**
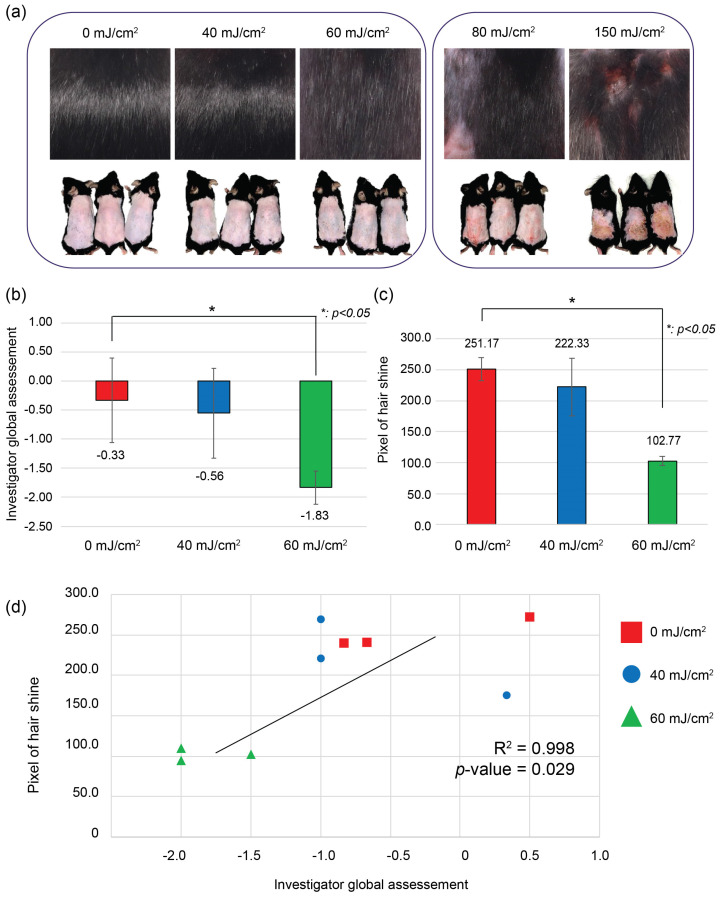
Changes of hair characteristics induced by ultraviolet (UV) irradiation: (**a**) Regrowth of rough hair after UVB irradiation of depilated skin. Deep burn inducing alopecic patches in doses of ≥80 mJ/cm^2^; (**b**) Hair luster loss evaluated by the Investigator’s Global Assessment (IGA) scale; (**c**) Loss of hair luster evaluated by pixel-based quantitative evaluation; (**d**) Correlation of pixel-based evaluation and the Investigator’s Global Assessment (IGA) scale. All experimental groups were conducted with *n* = 3.

**Figure 2 ijms-25-01885-f002:**
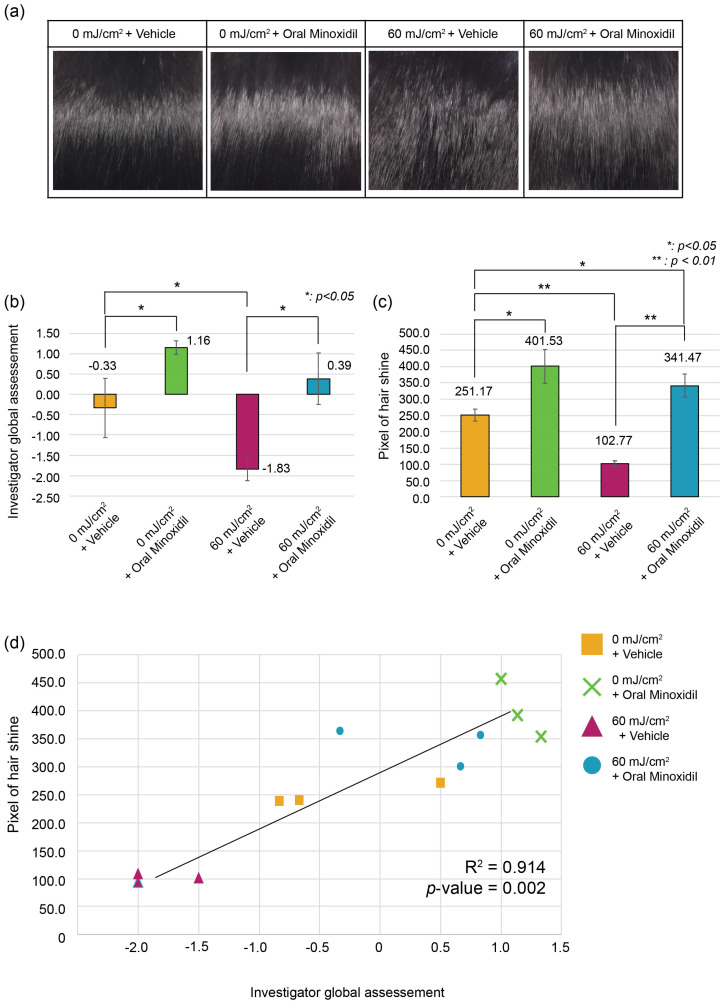
Reversibility of hair luster by oral minoxidil: (**a**) Hair luster improvement induced by oral minoxidil administration; (**b**) Minoxidil administration shows reversal of hair luster evaluated by the IGA scale in both 0 mJ/cm^2^, 60 mJ/cm^2^ irradiated groups; (**c**) Reversal of hair luster after minoxidil administration evaluated by pixel-based quantitative evaluation in both 0 mJ/cm^2^, 60 mJ/cm^2^ irradiated groups; (**d**) Correlation of pixel-based evaluation and the IGA scale. All experimental groups were conducted with *n* = 3.

**Figure 3 ijms-25-01885-f003:**
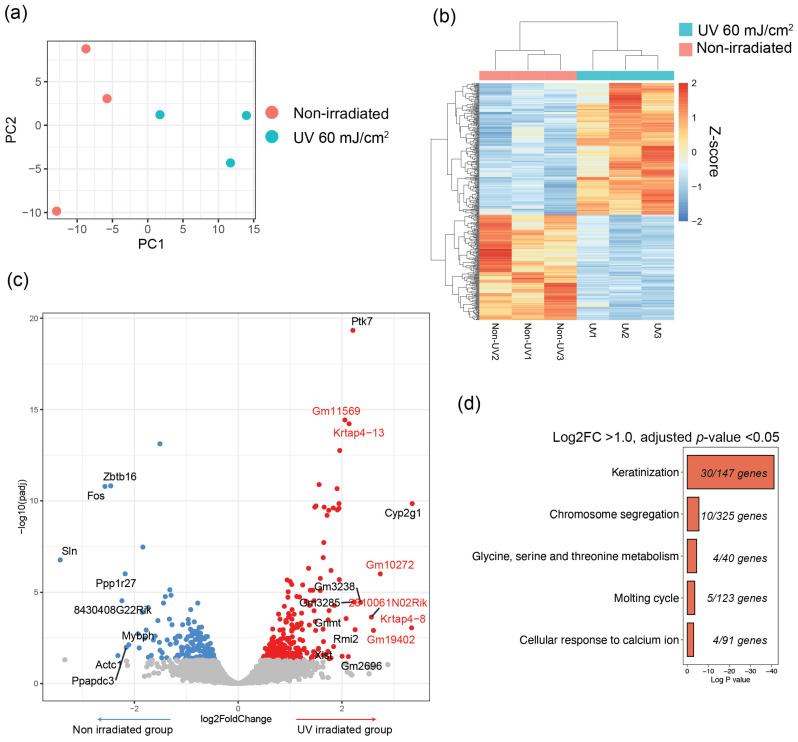
Transcriptomic differences between UV-irradiated and nonirradiated skin in C57BL/6 mice.: (**a**) Principal component analysis (PCA) plot of groups irradiated with 60 mJ/cm^2^ and 0 mJ/cm^2^; (**b**) Heatmap of transcriptomic differences of each group. *Y*-axis means gene cluster based on hierarchical clustering. The expression level (z-score) was visualized using color scale.; (**c**) Volcanoplot of differentially expressed genes between each group; Red dots are upregulated genes with adjusted *p*-value < 0.05 in an UV-irradiated group; Blue dots are downregulated genes with adjusted *p*-value < 0.05 in an UV-irradiated group. *X*-axis means log2 fold change of gene levels. *Y*-axis means -log10(adjusted *p*-value); (**d**) Pathway analysis of upregulated genes in the UV-irradiated group.

**Figure 4 ijms-25-01885-f004:**
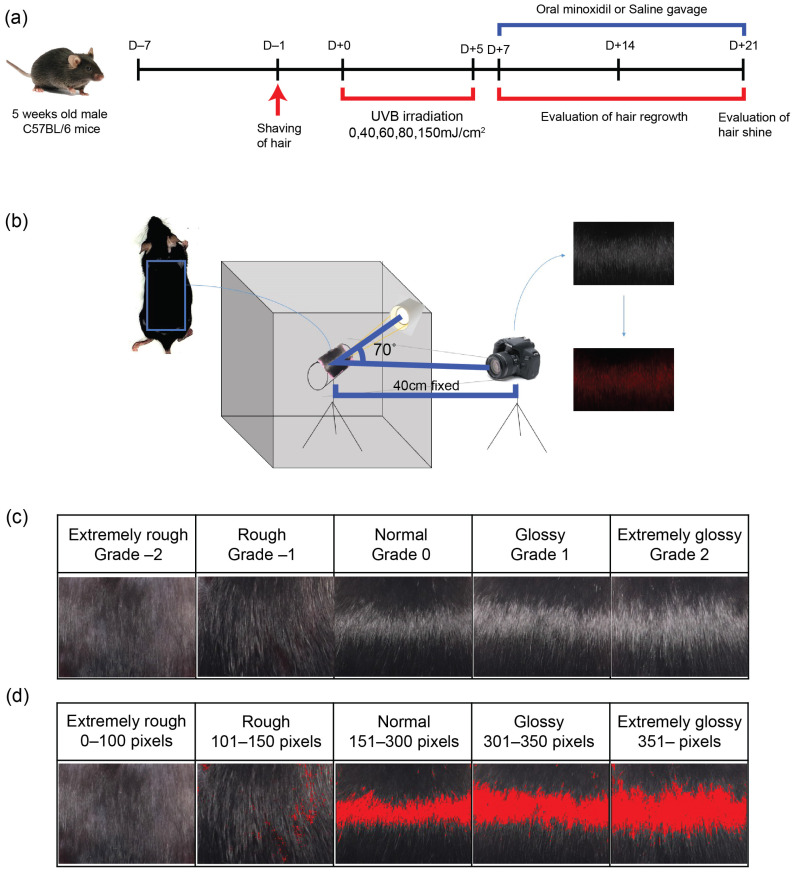
Animal model for inducing loss of hair gloss through UV irradiation: (**a**) Timeline schedule for induction of changes in hair luster; (**b**) Hair gloss assessment by illumination of a light source and photographic method; (**c**) Investigator’s global assessment (IGA) of hair luster; (**d**) Evaluation of hair luster using pixel-based method.

## Data Availability

The data supporting the findings of this study can be found in the Appendix A of this article. Data not included in the Appendix A are available from the corresponding author upon reasonable request.

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
