# Peer review of "Quantitative Analysis of Hair Luster in a Novel Ultraviolet-Irradiated Mouse Model"

_ijms, 2024, doi:10.3390/ijms25031885_

Round 1

Reviewer 1 Report

Comments and Suggestions for Authors

The description of the experimental method lacks precision, making it difficult to evaluate and replicate. Since the reliable replication of the experimental results for a standardised method is the main aim of the project, such matters acquire crucial importance.

Notably:

How many mice were in each group? (the photograph in Fig, 1 A suggests 4). What was the dispersion of the individual results and associated error margins? It is important to justify a four or five significant figures result in the reporting.

The total UV exposure energy is noted, but the intervals and the intensity are not stated, i.e. was the exposure continuous or intermittent. If it was intermittent, at what intervals was is  administered at what dose.

Were the mice allowed to move freely during the exposure?

In as similar manner, the angle of the camera and orientation of the mice need to be given, as this would have a material effect on the lustre evaluation.

Figure 3 B needs more explanation, particularly the significance of the unlabelled y-axis elements.

A specific reference for the IGA 5-point assessment tool should be provided.  

Author Response

<Reviewer #1>

The description of the experimental method lacks precision, making it difficult to evaluate and replicate. Since the reliable replication of the experimental results for a standardized method is the main aim of the project, such matters acquire crucial importance.

Ans) We appreciate your feedback on the importance of precise methodology. Accordingly, we have revised the relevant sections for clearer description and replication guidance.

How many mice were in each group? (the photograph in Fig, 1 A suggests 4). What was the dispersion of the individual results and associated error margins? It is important to justify four or five significant figures result in the reporting.

Ans) We have included 3 mice in each group for our study, focusing on the evaluation of UVB irradiation and the effect on hair luster. Additionally, another set of 3 mice per group were utilized for bulk RNA sequencing. Therefore, for clarification, we have revised Figure 1a to show 3 mice that were actually evaluated for the IGA and pixel-based analysis. Furthermore, we have included data results with standard deviation in the manuscript. This enhances the clarity in identifying significant results.

The total UV exposure energy is noted, but the intervals and the intensity are not stated, i.e. was the exposure continuous or intermittent. If it was intermittent, at what intervals was is administered at what dose.

Ans) Irradiation was administered daily for consecutive 5 days. The total irradiation dose was each 0, 200, 300, 400, 600mJ/cm². We have made clarifications on the manuscript (Methods section).

Were the mice allowed to move freely during the exposure?

Ans) During exposure, the mice were sedated with isoflurane (1.5%-2%) (N01AB06, Hana Pharm, Korea) for 1-2 minutes and exposed to UVB.

In a similar manner, the angle of the camera and orientation of the mice need to be given, as this would have a material effect on the lustre evaluation.

Ans) We acknowledge the importance of consistent camera positioning. We have made clarification for the angle and camera settings, light source. Also, we have made modifications for Figure 4b.

Figure 3B needs more explanation, particularly the significance of the unlabelled y-axis elements.

Ans) We have added more information in the Figure legend including y-axis information for Figure 3b.

A specific reference for the IGA 5-point assessment tool should be provided.  

Ans) To our best knowledge, we were unable to find any studies that used a simplified, visually-based luster measurement scale. Thus, this study introduces the novel application of the IGA point scale for assessing hair luster.

Reviewer 2 Report

Comments and Suggestions for Authors

The manuscript titled "Quantitative analysis of hair luster in a novel ultraviolet-irradiated mouse model “is an original research that has tried to introduce a new mouse model for hair-related traits.

The research presented has some merit because doing transcriptome analysis concerning hair characteristics and keratin proteins.

Please see my major comments as:

Introduction: The main comment is about the novelty part of the article. Authors should specify which part of the paper is new? I mean, are the materials and methods novel? Or is the effect of minoxidil on recovery after UV radiation in novel? Or, is the correlation between the IGA scale and the pixel-based assessment? There is ambiguity about what the authors' intention is.

Second main comment is related to incomplete Material and methods.

             Experimental design presented in incomplete structure. Please provide sufficient information about number of repeat per each treatment group.

             Statistical analysis model is absent. There is no P-values threshold.

             Many traits have been examined in terms of transcriptome, but unfortunately few phenotypic traits have been measured.

Do you measured other traits of hair, i.e., diameter of hairs,

Results: In your results, did you see the beneficial effect of minoxidil on which type of follicles or do you think it is? Do you have a microscopic photo of the hair scales? Is the effect of minoxidil related to the entire hair growth or to the scales of a specific group of hairs?

Discussion: Minoxidil is known to prolong anagen duration, and increase hair diameter.  But the role of minoxidil on luster characteristics will be interesting if it is well discussed. The discussion section is incomplete in that it does not discuss possible mechanisms of action of minoxidil. The stages of follicular growth under oral use of Minoxidil are not discussed.

What is the hypothesis of the researchers? Does the minoxidil cause the follicles to exit the telogen phase? How does this affect the luster?

Minor comments:

L45 what is the hypothesis concerning the effect of UV on hair growth.  Is it affect Telogen phase? Or anagene phase period?

Figure1: P-value of correlation? Number of samples? Number of investigators (operator)? Pearson or spearman correlation?

Figure 2: p value? Number of samples? Number of investigators (operator)?

Figure 2: Why you had not a treatment level with 0 mJ/cm2+minoxidil?

L106: … by hair fibers.  Please mention reference?

L191: Please provide statistical analysis. With a descriptive statistics about the measured traits. Please mention live body weight of animals +- SD in each treatment. Or maybe diameter of hair et ce …

Author Response

<Reviewer #2>

The manuscript titled "Quantitative analysis of hair luster in a novel ultraviolet-irradiated mouse model “is an original research that has tried to introduce a new mouse model for hair-related traits.

The research presented has some merit because doing transcriptome analysis concerning hair characteristics and keratin proteins.

Thank you very much for understanding the novelty of our research providing a novel mouse model in loss of hair luster.

Please see my major comments as:

Introduction: The main comment is about the novelty part of the article. Authors should specify which part of the paper is new? I mean, are the materials and methods novel? Or is the effect of minoxidil on recovery after UV radiation in novel? Or, is the correlation between the IGA scale and the pixel-based assessment? There is ambiguity about what the authors’ intention is.

Ans) We recognize that our initial presentation may have lacked clarity regarding the novelty of our work. To highlight, our study represents a significant advancement as it introduces the first animal model for studying hair luster. Although the methodological aspects are also new, we consider these evaluation methods as integral to the new model, serving as a disease model. In this background, the last sentences of the original introduction specify that we established a new animal model due to the absence of such animal models.

Second main comment is related to incomplete Material and methods.

Experimental design presented in incomplete structure. Please provide sufficient information about number of repeats per each treatment group.

Ans) We recognize the importance of clearly detailing our methodologies. We have added many details to the methodology section including mice numbers in the revised manuscript.

 Statistical analysis model is absent. There is no P-values threshold.

Ans) Statistical methodology is added in the methodology section. The added part of the manuscript is as follows;

 All experimental data were presented as a mean ± standard deviation or percentages, and all experiments were conducted at least in triplicates (n 3). Statistical analysis was performed using the SPSS Statistics 25.0 (IBM Corp., NY, United States) program. Statistical significance was calculated using the independent samples t-test. Statistical significance was considered at p < 0.05 and p < 0.01, indicated by * and ** respectively, depending on the group of comparison.

 Many traits have been examined in terms of transcriptome, but unfortunately few phenotypic trait have been measured. Do you measured other traits of hair, i.e., diameter of hairs,

Ans) As shown in Figure 3d, the majority of the changes in the transcriptome were related to genes associated with keratinization. This led us to focus on the luster phenotype, which was prominently observable. However, we acknowledge that we did not thoroughly examine other micro-phenotypes such as surface changes and diameter that can influence luster. Following your excellent suggestion, we believe that additional research using techniques like electron microscopy will be instrumental in further elucidating the factors that determine these phenotypic expressions. One of the factors that we had measured was hair growth rate, to evaluate whether UVB affects the general growth of hair, did not show any differences, as provided in supplementary data.

Results: In your results, did you see the beneficial effect of minoxidil on which type of follicles or do you think it is? Do you have a microscopic photo of the hair scales? Is the effect of minoxidil related to the entire hair growth or to the scales of a specific group of hairs?

Ans) As you correctly pointed out, incorporating histological analysis, including the examination of hair follicle subtypes, could be beneficial. While our current study did not encompass histological analysis or the examination of specific hair follicle subtypes in its scope, focusing primarily on luster as a specular reflection of light observable to the naked eye, we acknowledge that such an analysis could provide valuable insights into the mechanisms at play.

In response to your question about the effects of minoxidil, our study did not differentiate between the effects on various follicle types or provide microscopic images of hair scales. We concentrated on the overall visual luster and did not investigate the effect of minoxidil at the level of individual hair follicle subtypes or scales. However, based on your suggestion, we recognize the importance of microscopic examination and the evaluation of surface changes and other hair traits at a finer scale. In future studies, we plan to explore the specific impacts of minoxidil more comprehensively, using microscopic techniques. We appreciate your valuable input and have included a discussion of these limitations in our manuscript.

Discussion: Minoxidil is known to prolong anagen duration, and increase hair diameter.  But the role of minoxidil on luster characteristics will be interesting if it is well discussed. The discussion section is incomplete in that it does not discuss possible mechanisms of action of minoxidil. The stages of follicular growth under oral use of Minoxidil are not discussed.

What is the hypothesis of the researchers? Does the minoxidil cause the follicles to exit the telogen phase? How does this affect the luster?

Ans) Thank you for highlighting the significance of minoxidil in our study. We acknowledge that the relationship between minoxidil and hair luster is not well understood. While it is known that minoxidil can affect hair curliness or texture, suggesting an impact on hair shaft quality, the specific mechanisms remain unclear. Our data indicate that mice treated with minoxidil showed an increased hair growth rate by Day 14, which may suggest an induction of rapid hair growth or the anagen phase. However, the direct correlation between hair luster and the hair follicle cycle or growth rate is not well-defined. Our luster evaluations at Day 21 did not reveal differences in hair growth areas, leading us to assume a limited direct impact on hair luster.

Additionally, reports of changes in hair texture and curliness in patients treated with oral minoxidil imply that its effects extend beyond hair diameter and cycle, potentially influencing hair shaft properties as well. We have acknowledged these limitations and areas for future research in our revised manuscript.

Minor comments:

L45 what is the hypothesis concerning the effect of UV on hair growth.  Is it affect Telogen phase? Or anagene phase period?

Ans) The effect of UV on hair growth is diverse. Regarding the hair cycle, UVB is reported that it can induce early catagen and cause oxidative damage in human. [1, 2]However in mice, UVB is suggested to up-regulate the expression of the Wnt signaling pathway and prolong anagen and telogen phases in the hair follicles. [3]These data suggest that the direct effect of UV on hair growth is not clear.

After acclimatization, the mice typically reached 6-7 weeks of age, and in the case of BL6 mice shaved without any special manipulation, it is assumed that the experiments began during the 2nd telogen stage. In our experiments, while we only confirmed differences in initial hair growth rates due to minoxidil, we did not observe any variations in hair growth speed caused by UV exposure.

Figure1, Figure 2: P-value of correlation? Number of samples? Number of investigators (operator)? Pearson or spearman correlation?

Ans) P-value of correlation is added. Number of samples are total 9. Each 3 mice per dosages.

Figure 2: Why you had not a treatment level with 0 mJ/cm2+minoxidil?

 Ans) Thank you for pointing out 0 mJ/cm2+minoxidil. We have data on minoxidil + 0mJ/cm2. We initially excluded this group as it seemed distracting, but following recommendations, we have now included the results in the revised manuscript.

L106: … by hair fibers.  Please mention reference?

Ans) Reference added in the manuscript.

L191: Please provide statistical analysis. With a descriptive statistics about the measured traits. Please mention live body weight of animals +- SD in each treatment. Or maybe diameter of hair etc.

Ans) Statistical analysis added in manuscript. Live body weight of animals are not measured since no change in general health nor feeding habits occurred. Diameter for the hair is to be further evaluated in studies.

REFERENCE

  1. Dario, M. F.; Baby, A. R.; Velasco, M. V., Effects of solar radiation on hair and photoprotection. J Photochem Photobiol B 2015, 153, 240-6.
  2. Lu, Z.; Fischer, T. W.; Hasse, S.; Sugawara, K.; Kamenisch, Y.; Krengel, S.; Funk, W.; Berneburg, M.; Paus, R., Profiling the response of human hair follicles to ultraviolet radiation. J Invest Dermatol 2009, 129, (7), 1790-804.
  3. Zhai, X.; Gong, M.; Peng, Y.; Yang, D., Effects of UV Induced-Photoaging on the Hair Follicle Cycle of C57BL6/J Mice. Clin Cosmet Investig Dermatol 2021, 14, 527-539.

Round 2

Reviewer 1 Report

Comments and Suggestions for Authors

Much improved paper 

Author Response

Much improved paper.

Ans) We appreciate your feedback. Your comments and questions have been sincerely helpful in developing our manuscript.

Reviewer 2 Report

Comments and Suggestions for Authors

Please add your response to my question to the manuscript, besie the references:

" Ans) The effect of UV on hair growth is diverse. Regarding the hair cycle, UVB is reported that it can induce early catagen and cause oxidative damage in human. [1, 2]However in mice, UVB is suggested to up-regulate the expression of the Wnt signaling pathway and prolong anagen and telogen phases in the hair follicles. [3]These data suggest that the direct effect of UV on hair growth is not clear.

After acclimatization, the mice typically reached 6-7 weeks of age, and in the case of BL6 mice shaved without any special manipulation, it is assumed that the experiments began during the 2nd telogen stage. In our experiments, while we only confirmed differences in initial hair growth rates due to minoxidil, we did not observe any variations in hair growth speed caused by UV exposure."

  1. Dario, M. F.; Baby, A. R.; Velasco, M. V., Effects of solar radiation on hair and photoprotection. J Photochem Photobiol B 2015, 153, 240-6.
  2. Lu, Z.; Fischer, T. W.; Hasse, S.; Sugawara, K.; Kamenisch, Y.; Krengel, S.; Funk, W.; Berneburg, M.; Paus, R., Profiling the response of human hair follicles to ultraviolet radiation. J Invest Dermatol 2009, 129, (7), 1790-804.
  3. Zhai, X.; Gong, M.; Peng, Y.; Yang, D., Effects of UV Induced-Photoaging on the Hair Follicle Cycle of C57BL6/J Mice. Clin Cosmet Investig Dermatol 2021, 14, 527-539.

Author Response

<Reviewer #2>

Please add your response to my question to the manuscript, besie the references:

" Ans) The effect of UV on hair growth is diverse. Regarding the hair cycle, UVB is reported that it can induce early catagen and cause oxidative damage in human. [1, 2]However in mice, UVB is suggested to up-regulate the expression of the Wnt signaling pathway and prolong anagen and telogen phases in the hair follicles. [3]These data suggest that the direct effect of UV on hair growth is not clear.

After acclimatization, the mice typically reached 6-7 weeks of age, and in the case of BL6 mice shaved without any special manipulation, it is assumed that the experiments began during the 2nd telogen stage. In our experiments, while we only confirmed differences in initial hair growth rates due to minoxidil, we did not observe any variations in hair growth speed caused by UV exposure."

  1. Dario, M. F.; Baby, A. R.; Velasco, M. V., Effects of solar radiation on hair and photoprotection. J Photochem Photobiol B 2015, 153, 240-6.
  2. Lu, Z.; Fischer, T. W.; Hasse, S.; Sugawara, K.; Kamenisch, Y.; Krengel, S.; Funk, W.; Berneburg, M.; Paus, R., Profiling the response of human hair follicles to ultraviolet radiation. J Invest Dermatol 2009, 129, (7), 1790-804.
  3. Zhai, X.; Gong, M.; Peng, Y.; Yang, D., Effects of UV Induced-Photoaging on the Hair Follicle Cycle of C57BL6/J Mice. Clin Cosmet Investig Dermatol 2021, 14, 527-539.

Ans) Thank you for your comment. We have included the response already in the manuscript. Though somewhat the words have changed, the meaning of the sentence are consistent

It is included as

From Line : 153

Though the mechanism of UVB on hair shaft is diverse, the definitive cause of loss of hair luster by UVB in regrowing hair follicles is currently not known. Some studies have shown that in regard of hair cycle, UVB can induce early catagen and cause oxidative damage in human hair follicles. [6] However, in mice, UVB, up-regulated the expression of the Wnt signaling pathway and prolonged anagen and telogen phases. These data suggest that the direct effect of UV on hair growth is not clear.
